# Efficacy of Xanthine Oxidase Inhibitors in Lowering Serum Uric Acid in Chronic Kidney Disease: A Systematic Review and Meta-Analysis

**DOI:** 10.3390/jcm11092468

**Published:** 2022-04-27

**Authors:** Yoojin Lee, Jennifer Hwang, Shaan H. Desai, Xiaobai Li, Christopher Jenkins, Jeffrey B. Kopp, Cheryl A. Winkler, Sung Kweon Cho

**Affiliations:** 1Molecular Genetic Epidemiology Section, Basic Research Laboratory, Center for Cancer Research, National Cancer Institute, Frederick, MD 21702, USA; fatelist@gmail.com (Y.L.); shaan.h.desai@gmail.com (S.H.D.); winklerc@mail.nih.gov (C.A.W.); 2Department of Pharmacology, Ajou University School of Medicine, Suwon 16499, Korea; 3Department of Internal Medicine, The Hospital of Central Connecticut, New Britain, CT 06052, USA; jennifer.hwang.med@gmail.com; 4Howard University College of Medicine, Washington, DC 20059, USA; 5Biostatistics and Clinical Epidemiology Service, National Institutes of Health, Bethesda, MD 20814, USA; xiaobai.li@nih.gov; 6Department of Internal Medicine, University of Connecticut Health Center, Farmington, CT 06030, USA; cjenkins@uchc.edu; 7Kidney Disease Section, National Institute of Diabetes and Digestive and Kidney Diseases, National Institutes of Health, Bethesda, MD 20814, USA; jeffreyk@intra.niddk.nih.gov

**Keywords:** dose-proportional response, meta-analysis, urate-lowering therapeutics

## Abstract

Objective: Current guidelines for gout recommend a treat-to-target approach with serum uric acid (SUA). However, there is little evidence for the dose-dependent effects of urate-lowering therapy (ULT). Herein, we analyzed the reported SUA-lowering effect and SUA target achievement differences for various doses of xanthine oxidase inhibitors. Methods: Approved ULT drugs were selected from the FDA Drug Database. We included prospective randomized controlled trials of ULT drugs from ClinicalTrials.gov, articles published in the journal “Drugs”, and Embase, a literature database. A meta-analysis was performed to determine the ability of different ULT drugs and doses to lower and maintain a target SUA < 6 mg/dL. Results: We identified 35 trials including 8172 patients with a baseline SUA of 8.92 mg/dL. The allopurinol, febuxostat, and topiroxostat showed dose-proportional SUA-lowering responses. Compared with allopurinol 300 mg daily, febuxostat 80 mg daily and 120 mg daily more effectively maintained SUA < 6 mg/dL. Conclusion: Allopurinol, febuxostat, and topiroxostat showed dose-proportional ability to lower and achieve a target SUA < 6 mg/dL. Significance and Innovations. We showed dose-dependent SUA lowering effects of allopurinol, febuxostat, and topiroxostat. Febuxostat is effective at ULT compared to allopurinol and could be potentially offered as an alternative agent when patients (1) have CKD, (2) have the human leukocyte antigen HLA-B*5801 allele, and (3) become refractory to allopurinol. Gradual allopurinol dose increase with a lower starting dose is needed in CKD.

## 1. Introduction

Uric acid, the final metabolic product of purine degradation, is a known or potential biomarker for various pathological conditions. Hyperuricemia is defined as a serum uric acid (SUA) level of >6.8 mg/dL or 404.5 µmol/L at physiological pH [1] and can lead to the development of gout and nephrolithiasis. It also has been associated with insulin resistance, type 2 diabetes mellitus (T2DM), hypertension, chronic kidney disease (CKD), cardiovascular disease, and tumor lysis syndrome [1,2,3,4,5,6,7,8,9,10,11,12,13]. Obesity is a strong, modifiable risk factor for gout and hyperuricemia [14]. Increasing evidence suggests that insulin resistance is a key mediator of the association between obesity and hyperuricemia [10,13,14].

Gout occurs when highly immunogenic monosodium urate crystals [15] form in joints and soft tissue and is promoted when the SUA concentrations exceed solubility limits. These crystals produce painful attacks and debilitating bone erosion [16]. According to the 2015–2016 National Health and Nutrition Examination Survey, the prevalence of hyperuricemia and gout in the United States was 20.1% and 3.9%, respectively, and both increased in prevalence since the previous survey in 1988–1994 [17,18]. An increasing prevalence of gout has also been reported in other developed countries [19].

Urate-lowering therapy (ULT) is the mainstay of chronic gout management. Both the 2020 American College of Rheumatology (ACR) and 2016 European Alliance of Associations for Rheumatology (EULAR) guidelines recommend a treat-to-target management strategy in the absence of tophi (aggregates of monosodium urate crystals), aiming to reduce SUA to <6 mg/dL. EULAR further recommends a target of SUA of <5 mg/dL when tophi are present [20,21]. As treatment success is inversely related to baseline SUA [22], researchers should account for baseline SUA when examining the effectiveness of ULT, as variations in baseline SUA can result in different efficacies of ULT.

ULTs comprise xanthine oxidase inhibitors (XOIs), uricosuric agents, and recombinant uricase. The XOIs inhibit the enzyme xanthine oxidase that metabolizes hypoxanthine to uric acid and thereby reduces uric acid generation. The most commonly used XOI, allopurinol, is an analog of hypoxanthine and xanthine; it competes with these molecules in binding xanthine oxidase and inhibits the formation of uric acid [23]. Febuxostat non-selectively inhibits oxidized and reduced forms of xanthine oxidase [23]. It is a competitive inhibitor of the enzyme. Topiroxostat, a selective non-purine molecule, competitively inhibits xanthine oxidase [24]. Allopurinol and febuxostat are approved for use in the United States, whereas topiroxostat is only approved in Japan. The 2020 ACR and 2016 EULAR guidelines recommend low-dose allopurinol as a first-line treatment for gout [20,21]. Although febuxostat has better SUA-lowering efficacy than allopurinol, there are ongoing concerns regarding cardiovascular safety arising from the Cardiovascular Safety of Febuxostat and Allopurinol in Patients With Gout study [25]. However, more recent studies show no increased cardiovascular risk after febuxostat initiation compared with allopurinol [26,27].

Uricosuric drugs, unlike XOIs, prevent the reuptake of uric acid by the proximal convoluted tubule, primarily through inhibition of the human urate-1 transporter (URAT1). Uricosuric agents include probenecid, benzbromarone, and lesinurad.

Recombinant uricases convert uric acid into a more water-soluble compound, allantoin, for excretion in the urine. Pegloticase and rasburicase are recombinant uricases. Both are pegylated to increase plasma half-life by reducing renal excretion, and pegylation may also provide resistance to proteolysis and reduce antigenicity. Both agents are recommended for treatment-refractory gout based on their efficacy in lowering SUA and promoting the resolution of tophi [28].

Understanding the ability of ULT drugs to lower and maintain a target SUA is critical for clinicians to manage and treat hyperuricemia and gout appropriately. Several studies have demonstrated the efficacy of different ULT drugs across class and dose. Here we performed a systematic review and meta-analysis to determine the SUA-lowering effect and achievement difference (AD) in maintaining the target SUA for different ULT drugs at various doses, considering baseline SUA in hyperuricemic patients.

## 2. Materials and Methods

### 2.1. Data Collection Strategy

We reviewed approved drugs for hyperuricosemia listed in the United States Food and Drug Administration’s Approved Drug Product Database (https://www.accessdata.fda.gov/scripts/cder/daf/index.cfm, accessed 14 June 2019). We selected allopurinol, febuxostat, topiroxostat, benzbromarone, probenecid, lesinurad, rasburicase, and pegloticase for inclusion in this systematic review.

We included randomized prospective clinical trials (phase 1 to 4) of ULT drugs for the treatment of gout, hyperuricemia, or tumor lysis syndrome, regardless of the study masking method. Trials that involved dose titration intervals in step-up protocols were included by recording the extent of SUA-lowering after participants reached the maintenance dose for the time defined by each study protocol.

Records were collected from the following four sources (accessed 14 June 2019): ClinicalTrials.gov (accessed 14 June 2019, reviewed: *n* = 300 studies; included: *n* = 27), articles published in the journal Drugs (ISSN: 0012-6667) (reviewed: *n* = 236 studies; included: *n* = 19), Embase (embase.com, accessed 14 June 2019, reviewed: *n* = 732 studies; included: *n* = 34), and one publication [29] that was a reference in an included study. We used the following search terms: allopurinol, febuxostat, topiroxostat, gout, and hyperuricemia. The studies were included up to June 2019. In total, we included 81 studies.

### 2.2. Data Extraction and Imputation of Missing Values

Data were extracted from the journal article associated with each clinical trial or the ClinicalTrials.gov entry when a journal article was unavailable. Data were entered into a standardized data extraction form.

In cases of missing data, we calculated the percent SUA change from initial and final SUA levels ($\frac{\mathrm{FinalSUA}-\mathrm{InitialSUA}}{\mathrm{InitialSUA}}\times 100$), computed standard error (SE) if standard deviation (SD) was available ($\mathrm{SE}=\frac{\mathrm{SD}}{\sqrt{n}}$), and estimated SD if SE or 95% confidence intervals (CIs) were provided ($\mathrm{SD}=\frac{\sqrt{n}\times \left(\mathrm{Upper}95\%\mathrm{CI}-\mathrm{Lower}95\%\mathrm{CI}\right)}{3.92}$).

SD and SE were unavailable for 34 out of 42 (80.95%) allopurinol arms, 13 out of 34 (38.24%) febuxostat arms, 11 out of 21 (52%) topiroxostat arms, 15 out of 16 (94%) benzbromarone arms, and 3 out of 4 (75%) probenecid arms. For these studies, we estimated the drug-specific pooled variance of the SE ($\sqrt{\overset{n}{\underset{i=1}{\sum }}(\frac{{\mathrm{SE}}_{1}{}^{2}}{n_{1}}+\frac{{\mathrm{SE}}_{2}{}^{2}}{n_{2}}+\ldots +\frac{{\mathrm{SE}}_{n}{}^{2}}{n_{n}}}$) and imputed that value as the SE for each trial arm.

### 2.3. Quality Control Using Inclusion/Exclusion Criteria

#### 2.3.1. SUA-Lowering Analysis

In the SUA-lowering analysis, we excluded trial arms for any of the following six reasons: (1) SD or SE for percent SUA lowering were not reported, which are required for meta-regression; (2) we excluded placebo arms; (3) baseline SUA was within normal range (<6.8 mg/dL); (4) a dosing frequency other than once daily was used; (5) a dose titration scheme was used; or (6) the study did not define CKD.

After excluding 152 trial arms (64.68%; Figure 1 and Appendix A), 35 studies encompassing the following 83 trial arms were included: 29 arms for allopurinol, 4 arms for benzbromarone, 28 arms for febuxostat, 1 arm for lesinurad, 1 arm for probenecid, 16 arms for topiroxostat, and 4 arms for a combination of two ULT drugs (Appendix A). Only allopurinol, febuxostat, and topiroxostat were included in the final meta-analysis due to insufficient data for other ULT drugs.

#### 2.3.2. Achievement Difference Analysis

In the AD analysis, we selected trial arms based on 2020 ACR and 2016 EULAR guidelines [20,21]. As the 2020 ACR guidelines strongly recommend allopurinol as a first-line ULT, we included all trials using allopurinol 300 mg daily (the standard dose) as a control group [20]. To analyze studies targeting maintenance of SUA <6 mg/dL, we included the following 7 trial arms for febuxostat: 2 arms at 40 mg daily, 4 arms at 80 mg daily, and 1 arm at 120 mg daily.

## 3. Statistical Analysis

Previous studies of ULT drugs used the following different dosing frequencies: daily, twice daily, three times daily, or every four days. We chose to focus on daily doses, the most commonly used regimen, to harmonize the SUA-lowering effect of different ULT drugs. Thus, we excluded all other dosing frequencies.

In estimation of the dose-dependent SUA-lowering effect, we conducted a meta-analysis using the metareg command of *Stata*, version 16 (StataCorp; 2019, College Station, TX, USA: StataCorp LLC.). To account for the effect of baseline SUA on dose-dependent SUA-lowering outcomes, we selected only trials including subjects with an elevated baseline SUA of >6 mg/dL. We adjusted the analysis for covariates, including CKD diagnosis, sample size, length of follow-up, collection site, and proportion of male participants in each arm, as all studies included a preponderance of males. The proportion of male participants was transformed into a binary variable based on the following median values for trial arms (0.82 for allopurinol arms, 0.95 for febuxostat arms, and 0.91 for topiroxostat arms): 1 for more male-dominant arms and 0 for less male-dominant arms. This transformation was performed because the proportion of male participants was highly skewed, and this degraded model fit when it was introduced as a covariate. The meta-regression was performed only for allopurinol, febuxostat, and topiroxostat after excluding unqualified trial arms.

To compare AD among ULT drugs, we conducted a meta-analysis using the meta command of *Stata*, version 16 (StataCorp; 2019, College Station, TX, USA: StataCorp LLC.) model. For the analysis, we selected only arms, including subjects with a baseline SUA >6.8 mg/dL, which we defined as the upper limit of normal. In the model, we entered the numbers of participants who achieved and did not achieve the intended therapeutic outcome, which yielded the efficacy of the intended therapeutic effect. Systematic bias and heterogeneity among studies are shown as a funnel plot (Appendix A).

## 4. Results

### 4.1. SUA-Lowering Analysis

#### 4.1.1. Trial Characteristics

The SUA-lowering meta-analysis included 8172 patients with a baseline SUA of 8.92 ± 0.77 mg/dL (mean ± SD; range, 6.88–10.40 mg/dL) after imputation. There were 3465 patients receiving allopurinol with a baseline SUA of 8.82 ± 0.74 mg/dL, 3899 patients on febuxostat with a baseline SUA of 9.25 ± 0.72 mg/dL, 506 patients on topiroxostat with a baseline SUA of 8.66 ± 0.55 mg/dL, 93 patients on benzbromarone with baseline SUA of 8.74 ± 0.34 mg/dL, 35 patients on probenecid with a baseline SUA of 9.07 ± 1.2 mg/dL, and 67 patients on a combination of ULT drugs with a baseline SUA of 8.35 ± 1.75 mg/dL.

#### 4.1.2. Meta-Regression

We conducted random-effects meta-regression to estimate the extent of SUA-lowering based on the dose of three xanthine oxidase inhibitors, allopurinol, febuxostat, and topiroxostat. Allopurinol (*n* = 29) dose was positively related to the extent of SUA-lowering (*β* = 0.05, SE = 0.01, *p* < 0.001, 95% CI 0.036, 0.072) (Table 1 and Appendix A). In the covariate model (*n* = 19), the extent of SUA-lowering was significantly related to allopurinol dose (*β* = 0.36, SE = 0.07, *p* = 0.001, 95% CI 0.19, 0.52) and marginally significantly related to CKD diagnosis (*β* = 27.62, SE = 12.71, *p* = 0.062, 95% CI −1.698, 56.942) (Table 2).

Febuxostat (*n* = 28) dose was positively related to the extent of serum uric acid-lowering (*β* = 0.18, SE = 0.03, *p* < 0.001, 95% CI 0.124, 0.230) (Table 1 and Appendix A). The covariate model (*n* = 25) also showed that the extent of SUA-lowering was significantly related to febuxostat dose (*β* = 0.23, SE = 0.04, *p* < 0.001, 95% CI 0.151, 0.301) (Table 3).

The Topiroxostat (*n* = 16) dose was also positively related to the extent of SUA-lowering (*β* = 0.10, SE = 0.02, *p* < 0.001, 95% CI 0.059, 0.141) (Table 1 and Appendix A). In the covariate model (*n* = 16), the extent of SUA-lowering was significantly related to the topiroxostat dose (*β* = 0.13, SE = 0.04, *p* = 0.007, 95% CI 0.044, 0.220) and proportion of male participants (*β* = −14.22, SE = 5.16, *p* = 0.020, 95% CI −25.720, −2.713) (Table 4). Furthermore, a post-hoc analysis showed that the topiroxostat dose was marginally significantly related to the extent of SUA-lowering among trial arms with a greater male predominance (*n* = 8, *β* = 0.11, SE = 0.04, *p* = 0.058, 95% CI −0.006, 0.217) but not among trial arms with a lesser male predominance (*n* = 8, *β* = 0.03, SE = 0.07, *p* = 0.581, 95% CI −0.171, 0.241).

### 4.2. Achievement Difference Analysis

#### 4.2.1. Trial Characteristics

We used achievement difference (AD) meta-analysis to examine which study arms achieved the target serum uric acid level, as defined in each study. This analysis included 1849 patients with a baseline SUA of 9.79 ± 0.18 mg/dL (mean ± SD; range, 9.43–9.98 mg/dL). There were 632 patients on allopurinol with a baseline SUA of 9.83 ± 0.13 mg/dL and 1217 patients on febuxostat with a baseline SUA of 9.76 ± 0.21 mg/dL.

#### 4.2.2. Meta-Analysis

Two trial arms with febuxostat 40 mg did not show a significant AD in maintaining SUA < 6 mg/dL when compared with allopurinol 300 mg daily (AD = 0.05, 95% CI −0.018, 0.109) (Table 5 and Figure 2). However, four arms with febuxostat 80 mg (AD = 0.25, 95% CI 0.165, 0.325) and one arm with febuxostat 120 mg (AD = 0.40, 95% CI 0.326, 0.483) showed significant AD outcomes compared with allopurinol 300 mg daily.

## 5. Discussion

Increased uric acid levels play a central role in the pathogenesis of particular diseases, including gout and tumor lysis syndrome, and are associated with a higher risk of cardiovascular disease [8,9], CKD [11], diabetes mellitus [10], and nonalcoholic fatty liver disease [10]. However, there have been few systematic reviews and meta-analyses of ULT. The present meta-analysis provides insight into the dose effects of different ULT drugs and may help guide clinicians in selecting the appropriate dosing to achieve a specific SUA-lowering effect.

We analyzed the following three different XOIs that showed dose-dependent SUA-lowering effects: allopurinol, febuxostat, and topiroxostat. These agents showed meaningful SUA-lowering effects that persisted after controlling for the presence of CKD, sample size, length of follow-up, collection site, and proportion of male participants. With regards to CKD, concerns have arisen with regard to using allopurinol in CKD due to its potentially detrimental effect on renal function, and untreated hyperuricemia may promote poor outcomes of gout or hyperuricemia [65]. Stamp et al. [66,67] studies were supported by results showing allopurinol dose escalation in CKD can achieve effective SUA with comparable safety; a lower starting dosing was required to achieve target SUA in CrCl < 30 mL/min. The analysis presented here shows CKD has a marginal influence on the SUA-lowering effects of allopurinol and hence supports the current ACR 2020 practice guidelines, which recommend a lower allopurinol starting dose in CKD [20].

The present meta-analysis showed that febuxostat had better SUA-lowering ability than allopurinol or topiroxostat (Table 1). Based on the null model, the expected SUA-lowering effect of allopurinol 300 mg was a 15% reduction from the baseline SUA level. Based on our SUA-lowering analysis, an equivalent outcome would be achieved by febuxostat 83.33 mg or topiroxostat 150 mg (Table 2, Table 3 and Table 4). Previous meta-analyses that examined the comparative efficacy of ULT drugs showed similar AD results as in the present study. Ye et al. [68] report that the efficacy of febuxostat 40 mg daily is higher than that of allopurinol 100–300 mg daily and is greater with higher doses (80–120 mg daily). Borghi et al. [69] showed that doses of febuxostat >40 mg daily are superior to allopurinol at 300 mg daily. Furthermore, Li et al. [70] suggested that febuxostat has greater efficacy at 120 mg daily compared to allopurinol at 300 mg daily. Based on our AD analysis results (Table 5), febuxostat 80 mg daily may be a better treatment option than allopurinol for patients with refractory SUA levels, consistent with 2016 EULAR and 2020 ACR recommendations [20,21].

The meta-analysis presented here suggests that febuxostat could be preferable to allopurinol for treating gout. Given the higher cost and black box warning associated with febuxostat, allopurinol is currently recommended as the first-line treatment for most gout patients. However, studies have shown the cost-effectiveness of febuxostat compared with allopurinol in chronic gout management [71,72]. Thus, it may be worthwhile to further examine the advantages of febuxostat from an economic perspective. Moreover, recent studies show no increased cardiovascular risk of febuxostat compared with allopurinol [26,27]. Apart from its overall greater efficacy, febuxostat may also be a better ULT option in specific instances. First, febuxostat may be more beneficial to patients with mild to moderate renal impairment (creatinine clearance of 30–80 mL/min/1.73 m^2^), as it is not influenced by renal function [73]. Second, febuxostat may be better for treating patients with the human leukocyte antigen HLA-B*5801 allele [74,75]. Previous studies reveal that allopurinol can cause a spectrum of severe skin reactions, including Stevens-Johnson syndrome and toxic epidermal necrolysis, in patients with the HLA-B*5801 allele [74,75]. Although rare, allopurinol hypersensitivity syndrome can cause serious mortality and morbidity. Therefore, according to ACR 2020 gout guidelines, testing for the HLA–B*5801 allele prior to starting allopurinol is recommended for patients of Southeast Asian descent and for African American patients [20]. Third, febuxostat may be a better choice for patients who are refractory to allopurinol. Individuals with a minor allele (rs2231142) of the ATP-binding cassette ABCG2 transporter show minimal response to allopurinol. Therefore, patients with a higher risk of treatment failure and a lack of biological sensitivity to allopurinol could consider initiating treatment with febuxostat [76].

Strengths of the present study include the analysis of SUA-lowering and AD efficacy of ULT drugs at different doses compared with allopurinol 300 mg daily, which is most commonly used in the United States [77]. We also included trials from various geographic regions, including the United States, Europe, China, Japan, Iran, and India. However, this analysis also has limitations, including (1) the specific details of the studies were not captured due to the nature of the meta-analysis technique; (2) heterogeneity among studies may have resulted from the inclusion of older studies between the 1970s and 1980s, which may have included patients with different lifestyles and employed different trial designs; (3) treatment duration and patient characteristics varied among trials; (4) we did not analyze the potential influence of renal function on treatment efficacy by accounting for patients’ body mass index, sex, or ethnicity; (5) there was a lack of standardized reporting of SUA-lowering effects among included trials.

We recommend that future studies analyze sex differences in hyperuricemia. Although men, compared to women, have a higher risk of developing gout (5.2% vs. 2.7%) and hyperuricemia (20.2% vs. 4.2%) [18], there is evidence that during the late menopausal transition stage, SUA rises in women to levels comparable to those of men at a similar age [78]. In addition, although body mass index is typically considered in clinical trials, future trials involving agents that affect SUA should report detailed information on the body mass index of participants, so that clinicians can better understand appropriate doses of ULT. Furthermore, standardized approaches to reporting data should be considered in future studies to improve the feasibility of subsequent meta-analyses.

In summary, the meta-analysis presented here shows the dose-dependent SUA-lowering effects of allopurinol, febuxostat, and topiroxostat. Further, it suggests that febuxostat is a more effective ULT than allopurinol, especially at higher doses. These findings support the ULT recommendations in the 2016 EULAR and 2020 ACR guidelines.

## Figures and Tables

**Figure 1 jcm-11-02468-f001:**
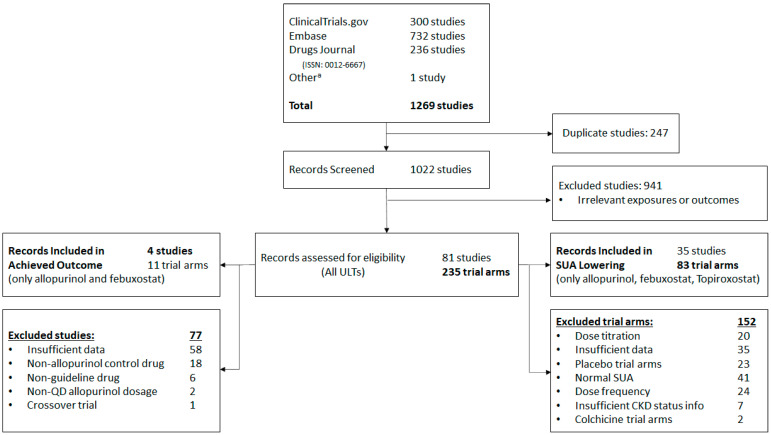
Flow diagram. Summary of literature search and selection process. *Note*. CKD: chronic kidney disease; QD: quarter in die (four times a day); SUA: serum uric acid; and ULT: uric lowering therapy. ^a^ One study was listed as a reference for an article sourced through Embase [27].

**Figure 2 jcm-11-02468-f002:**
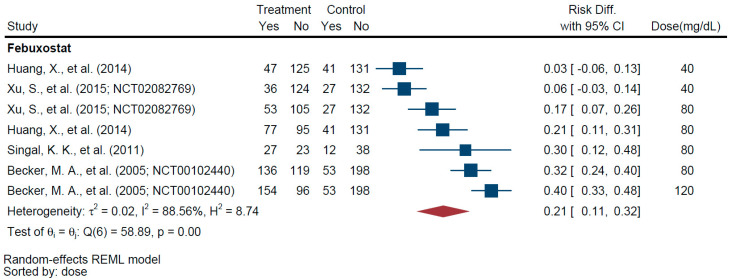
Forest Plot of Achievement Difference in Febuxostat Monotherapy vs. Allopurinol 300 mg QD–SUA < 6 mg/dL [30,31,35,64].

**Table 1 jcm-11-02468-t001:** Meta-analysis of Serum uric acid-lowering effect of approved uric lowering therapy drugs with daily doses [29,30,31,32,33,34,35,36,37,38,39,40,41,42,43,44,45,46,47,48,49,50,51,52,53,54,55,56,57,58,59,60,61,62,63].

ULT Drugs									
	Allopurinol			Febuxostat			Topiroxostat		
	*β*	SE	*p*	*β*	SE	*p*	*β*	SE	*p*
n	29			28			16		
τ^2^	26.35			24.03			17.25		
I^2^ residual (%)	91.93			85.09			89.03		
Adjusted R^2^	0.60			0.70			0.67		
Intercept	17.52	2.03	<0.001	31.97	2.11	<0.001	23.31	1.88	<0.001
Dose	0.05	0.01	<0.001	0.18	0.03	<0.001	0.10	0.02	<0.001
Dose Range	50–435			20–240			20–152		

Legend. Shown are the beta values for sample size and R^2^ (the proportion of the variation in the dependent variable that is explained by the independent variable), residual, and R^2^ adjusted for chronic kidney disease status, sample size, length of follow-up, data collection site, and proportion of male participants. For the Y intercept and dose, beta coefficient, and standard errors of the intercept are also shown. Also shown are the range of typical daily doses for each drug.

**Table 2 jcm-11-02468-t002:** Covariate model for allopurinol and post-hoc analyses based on data collection site [29,30,31,32,33,34,35,36,37,38,39,40,41,42,43,44,45,46,47,48,49,50,51,52,53,54,55,56,57,58,59,60,61,62,63].

Reference Collection Site	Middle East			Multi-National			North America		
	*β*	SE	*p*	*β*	SE	*p*	*β*	SE	*p*
Intercept	−46.36	20.18	0.051	−109.20	32.54	0.010	−102.39	32.47	0.014
Dose	0.36	0.07	0.001	0.36	0.07	0.001	0.36	0.07	0.001
Chronic kidney disease (CKD) diagnosis	27.62	12.71	0.062	27.62	12.71	0.062	27.62	12.71	0.062
Sample size	0.002	0.01	0.771	0.002	0.01	0.771	0.002	0.01	0.771
Follow-up (days)	0.03	0.02	0.090	0.03	0.02	0.090	0.03	0.02	0.090
Region									
China	−26.39	4.91	0.001	36.45	11.98	0.016	29.65	10.99	0.027
Europe	−109.98	23.92	0.002	−47.14	14.28	0.011	−53.95	11.84	0.002
Japan	15.91	5.61	0.022	78.75	18.55	0.003	71.95	17.57	0.003
Middle Eastern	-	-	-	62.84	14.10	0.002	56.03	12.83	0.002
Multi-National	−62.84	14.10	0.002	-	-	-	-6.81	6.21	0.305
North America	−56.03	12.83	0.002	6.84	6.21	0.305	-	-	-
Proportion of males	−8.42	6.49	0.231	−8.42	6.49	0.231	−8.42	6.49	0.231

Legend. The covariate model used each clinical trial location as a reference group: *n* = 19, τ^2^ = 7.32, I^2^ residual (%) = 76.74, adjusted R^2^ = 0.91. The dose-dependent serum uric acid-lowering effect was estimated while accounting for CKD status, sample size, length of follow-up, data collection site, and proportion of male participants. CKD was defined as eGFR < 60 mL/min/1.73 m^2^, CrCl < 50 mL/min/1.73 m^2^, or serum creatinine > 1.5 mg/dL. The proportion of male participants was categorized by the median score for all arms: 0 for less male-dominant arms and 1 for more male-dominant arms. Region refers to the site where clinical trials were conducted. Follow-up refers to the duration of the clinical trial.

**Table 3 jcm-11-02468-t003:** Covariate model for febuxostat and post-hoc analyses based on data collection site [29,30,31,32,33,34,35,36,37,38,39,40,41,42,43,44,45,46,47,48,49,50,51,52,53,54,55,56,57,58,59,60,61,62,63].

Reference Collection Site	China			Europe			Japan			North America		
	*β*	SE	*p*	*β*	SE	*p*	*β*	SE	*p*	*β*	SE	*p*
Intercept	18.85	6.94	0.015	23.23	7.74	0.008	30.68	3.90	<0.001	28.14	4.29	<0.001
Dose	0.23	0.04	<0.001	0.23	0.04	<0.001	0.23	0.04	<0.001	0.23	0.04	<0.001
Chronic kidney disease (CKD) diagnosis	−9.67	9.05	0.301	−9.67	9.05	0.301	−9.67	9.05	0.301	−9.67	9.05	0.301
Sample size	0.003	0.01	0.801	0.003	0.01	0.801	0.003	0.01	0.801	0.003	0.01	0.801
Duration (days)	0.02	0.03	0.492	0.02	0.03	0.492	0.02	0.03	0.492	0.02	0.03	0.492
Region												
China	-	-	-	−4.38	9.71	0.658	−11.83	5.02	0.032	−9.29	5.82	0.130
Europe	4.38	9.71	0.658	-	-	-	−7.45	8.61	0.400	−4.91	7.84	0.540
Japan	11.83	5.02	0.032	7.45	8.61	0.400	-	-	-	2.54	3.61	0.492
North America	9.29	5.82	0.130	4.91	7.84	0.540	−2.54	3.61	0.492	-	-	-
Male proportion	0.26	3.07	0.933	0.26	3.07	0.933	0.26	3.07	0.933	0.26	3.07	0.933

Legend: The covariate model used each clinical trial location as a reference group: *n* = 25, τ^2^ = 15.55, I^2^ residual (%) = 71.84, adjusted R^2^ = 0.81. The dose-dependent serum uric acid-lowering effect was estimated while accounting for CKD status, sample size, length of follow-up, data collection site, and proportion of male participants. CKD was defined as eGFR < 60 mL/min/1.73 m^2^, creatinine clearance < 50 mL/min/1.73 m^2^ [correct?), or serum creatinine > 1.5 mg/dL. The proportion of male participants was categorized by the median score for all arms: 0 for less male-dominant arms and 1 for more male-dominant arms. Region refers to the site where clinical trials were conducted. Duration refers to the length of the clinical trial.

**Table 4 jcm-11-02468-t004:** Covariate model for topiroxostat and post-hoc analyses based on proportion of male participants [29,30,31,32,33,34,35,36,37,38,39,40,41,42,43,44,45,46,47,48,49,50,51,52,53,54,55,56,57,58,59,60,61,62,63].

	Japan			More Male-Dominant			Less Male-Dominant		
	*β*	SE	*p*	*β*	SE	*p*	*β*	SE	*p*
*n*	16			8			8		
τ^2^	10.57			1.71			2.81		
I^2^ residual (%)	82.55			34.25			70.03		
Adjusted R^2^	80.06			97.07			94.73		
Intercept	30.23	10.57	0.017	−34.53	15.73	0.117	21.73	7.18	0.039
Dose	0.12	0.03	0.003	0.11	0.04	0.058	0.03	0.06	0.581
Chronic kidney disease (CKD) diagnosis	−10.93	5.70	0.084	−23.13	4.35	0.013			
Sample size	−0.008	0.06	0.902	−0.08	0.04	0.172	0.28	0.06	0.231
Follow-up (days)	1.09	2.80	0.705	13.91	4.45	0.052	0.11	1.80	0.954
Male proportion	−14.22	5.16	0.020						

Legend: The dose-dependent serum uric acid-lowering effect was estimated while accounting for CKD status, sample size, length of follow-up, data collection site, and proportion of male participants. CKD was defined as eGFR < 60 mL/min/1.73 m^2^, CrCl < 50 mL/min/1.73 m^2^, or serum creatinine > 1.5 mg/dL. The proportion of male participants was categorized by the median score for all arms: 0 for less male-dominant arms and 1 for more male-dominant arms. Region refers to the site where clinical trials were conducted. Follow-up refers to the length of the clinical trial.

**Table 5 jcm-11-02468-t005:** Serum uric acid-lowering Achievement Difference vs. allopurinol 300 mg daily.

Study (Year; NCT)	Achieved Outcome Goal	Drug/Dose (mg)	AD vs. Allopurinol 300 mg Daily (95% CI)	AD vs. Allopurinol 300 mg Daily by Dose (95% CI)
Xu et al. (2015; NCT02082769) [30]	SUA < 6 mg/dL at last three visits (6 months of treatment) **	Febuxostat 40 mg daily	0.06 (−0.032, 0.142)	0.05 (−0.018, 0.109)
Huang et al. (2014) [35]	SUA < 6 mg/dL at last three monthly visits (7 months of treatment)	Febuxostat 40 mg daily	0.04 (−0.057, 0.127)	
Xu et al. (2015; NCT02082769) [30]	SUA < 6 mg/dL at last three visits (6 months of treatment) **	Febuxostat 80 mg daily	0.17 (0.072, 0.260) *	0.25 (0.165, 0.325) *
Singal et al. (2011) [64]	SUA < 6 mg/dL at last three measurements (6 months of treatment)	Febuxostat 80 mg daily	0.30 (0.118, 0.482) *	
Huang et al. (2014) [35]	SUA < 6 mg/dL at last three monthly visits (7 months of treatment)	Febuxostat 80 mg daily	0.21 (0.111, 0.307) *	
Becker et al. (2005; NCT00102440) [31]	SUA < 6 mg/dL at last three monthly visits (1 year of treatment)	Febuxostat 80 mg daily	0.32 (0.243, 0.402) *	
Becker et al. (2005; NCT00102440) [31]	SUA < 6 mg/dL at last three monthly visits (1 year of treatment)	Febuxostat 120 mg daily	0.40 (0.326, 0.483) *	0.40 (0.326, 0.483) *

Legend: * statistically significant; ** converted from 356.9 µmol/L using 59.48 µmol/L = 1 mg/dL. NCT: the national clinical trial number.

## Data Availability

Raw data are available at https://github.com/fatelist82/Gout-Therapeutics (accessed on 14 March 2022).

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
