# Peer review of "Efficacy of Xanthine Oxidase Inhibitors in Lowering Serum Uric Acid in Chronic Kidney Disease: A Systematic Review and Meta-Analysis"

_jcm, 2022, doi:10.3390/jcm11092468_

Round 1

Reviewer 1 Report

This was a nice meta analysis which compared effecacy of urate lowering therapies,  specifically allopurinol, febuxostat, and topiroxostat. This study found that febuxostat was more effective when used in hyperuricemia, gout, and tumor lysis syndrome.

Overall a good review of the drug mechanism of actions as well as the studies on efficacy as a whole. The tables were clear and not difficult to read. It would be nice to include nephrolithiasis as a sequela of hyperuricemia as well- what about obesity as an independent risk factor for hyperuricemia- might be worth a mention especially since because obesity has become worse it is attributing to further hyperuricemia in the US as well. 

Overall great work

Author Response

Comment #1: It would be nice to include nephrolithiasis as a sequela of hyperuricemia as well.

  • Response #1:

As hyperuricemia is traditionally associated with gout and nephrolithiasis, I have mentioned the following concept in the introduction (p. 1-2 line 39-43).

“Hyperuricemia is defined as a serum uric acid (SUA) level >6.8 mg/dL or 404.5 µmol/L at physiological pH1, and can lead to development of gout and nephrolithiasis.”

Comment #2: What about obesity as an independent risk factor for hyperuricemia- might be worth a mention especially since because obesity has become worse it is attributing to further hyperuricemia in the US as well.

  • Response #2:

We edited the insight related to the link between obesity and hyperuricemia (p. 2 line 44-46).

“Obesity is a strong modifiable risk for gout and hyperuricemia2. Increasing evidence suggests insulin resistance may be a key mediator between obesity and hyperuricemia as both prevalence and incidence of obesity and gout are increasing2,3,4.”

Reviewer 2 Report

I consider this to be an interesting study to clarify the use of the most appropriate therapy in hyperuricemia  with xanthine oxidase inhibitors.

in the abstract spell the word SUA.

correct in point 4, resultats by results.

DISCUSSION: it could be useful to determine in what circumstances it would be advisable to indicate the determination of human leukocyte antigen HLA-B*5801 allele

Author Response

Comment #1: In the abstract, spell the word SUA.

  • Response #1:

We have inserted the full spelling of SUA in the abstract (p.1 line 18-19).

Comment #2: Correct in point 4, resultats by results.

  • Response #2:

We have corrected the spelling error in Results section.

Comment #3: It could be useful to determine in what circumstances it would be advisable to indicate the determination of human leukocyte antigen HLA-B*5801 allele.

  • Response #3:

We have added the specific recommendations regarding HLA-B*5801 allele testing as we agree that it is important to readers to know when to test or know HLA status (p. 8 line 289-292).

“Therefore according to ACR 2020 gout guidelines, testing for the HLA–B*5801 allele prior to starting allopurinol is recommended for patients of Southeast Asian descent and for African American patients5
